# ATTENTIVE NEURAL PROCESSES

**Hyunjik Kim**[1,2*], **Andriy Mnih**[1], **Jonathan Schwarz**[1], **Marta Garnelo**[1], **Ali Eslami**[1],
**Dan Rosenbaum**[1], **Oriol Vinyals**[1], **Yee Whye Teh**[1,2]
DeepMind[1], University of Oxford[2]

## ABSTRACT

Neural Processes (NPs) (Garnelo et al., 2018a;b) approach regression by learning to map a context set of observed input-output pairs to a distribution over regression functions. Each function models the distribution of the output given an input, conditioned on the context. NPs have the benefit of fitting observed data efficiently with linear complexity in the number of context input-output pairs, and can learn a wide family of conditional distributions; they learn predictive distributions conditioned on context sets of arbitrary size. Nonetheless, we show that NPs suffer a fundamental drawback of underfitting, giving inaccurate predictions at the inputs of the observed data they condition on. We address this issue by incorporating attention into NPs, allowing each input location to attend to the relevant context points for the prediction. We show that this greatly improves the accuracy of predictions, results in noticeably faster training, and expands the range of functions that can be modelled.

## 1 INTRODUCTION

Regression tasks are usually cast as modelling the distribution of a vector-valued output $y$ given a vector-valued input $x$ via a deterministic function, such as a neural network, taking $x$ as an input. In this setting, the model is trained on a dataset of input-output pairs, and predictions of the outputs are independent of each other given the inputs. An alternative approach to regression involves using the training data to compute a *distribution over functions* that map inputs to outputs, and using draws from that distribution to make predictions on test inputs. This approach allows for reasoning about multiple functions consistent with the data, and can capture the co-variability in outputs given inputs. In the Bayesian machine learning literature, non-parametric models such as Gaussian Processes (GPs) are popular choices of this approach.

Neural Processes (NPs) (Garnelo et al., 2018a;b) offer an efficient method to modelling a distribution over regression functions, with prediction complexity linear in the context set size. Once trained, they can predict the distribution of an arbitrary *target* output conditioned on a set of *context* input-output pairs of an arbitrary size. This flexibility of NPs enables them to model data that can be interpreted as being generated from a stochastic process. It is important to note however that NPs and GPs have different training regimes. NPs are trained on samples from multiple realisations of a stochastic process (i.e. trained on many different functions), whereas GPs are usually trained on observations from one realisation of the stochastic process (a single function). Hence a direct comparison between the two is usually not plausible.

Despite their many appealing properties, one substantial weakness of NPs is that they tend to underfit the context set. This manifests in the 1D curve fitting example on the left half of Figure 1 as inaccurate predictive means and overestimated variances at the input locations of the context set. The right half of the figure shows this phenomenon when predicting the bottom half of a face image from its top half: although the prediction is globally coherent, the model's reconstruction of the top-half is far from perfect. In an NP, the encoder aggregates the context set to a fixed-length latent summary via a permutation invariant function, and the decoder maps the latent and target input to the target output. We hypothesise that the underfitting behaviour is because the mean-aggregation step in the encoder acts as a bottleneck: since taking the mean across context representations gives the same weight to each context point, it is difficult for the decoder to learn which context points

---
*Corresponding author: `hyunjikk@google.com`

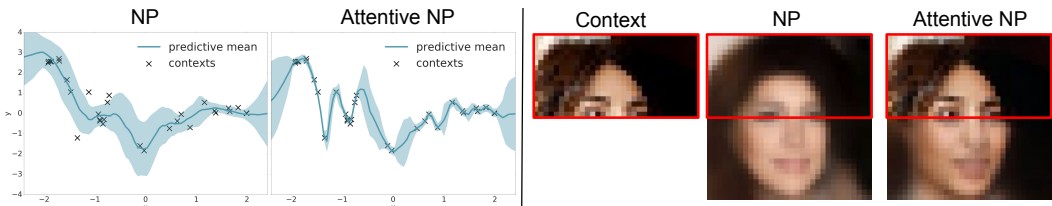

Figure 1: Comparison of predictions given by a fully trained NP and Attentive NP (ANP) in 1D function regression (left) / 2D image regression (right). The contexts (crosses/top half pixels) are used to predict the target outputs ($y$-values of all $x \in [-2, 2]$/all pixels in image). The ANP predictions are noticeably more accurate than for NP at the context points.

provide relevant information for a given target prediction. In theory, increasing the dimensionality of the representation could address this issue, but we show in Section 4 that in practice, this is not sufficient.

To address this issue, we draw inspiration from GPs, which also define a family of conditional distributions for regression. In GPs, the kernel can be interpreted as a measure of similarity among two points in the input domain, and shows which context points $(x_i, y_i)$ are relevant for a given query $x_*$. Hence when $x_*$ is close to some $x_i$, its $y$-value prediction $y_*$ is necessarily close to $y_i$ (assuming small likelihood noise), and there is no risk of underfitting. We implement a similar mechanism in NPs using differentiable attention that learns to attend to the contexts relevant to the given target, while preserving the permutation invariance in the contexts. We evaluate the resulting *Attentive Neural Processes* (ANPs) on 1D function regression and on 2D image regression. Our results show that ANPs greatly improve upon NPs in terms of reconstruction of contexts as well as speed of training, both against iterations and wall clock time. We also demonstrate that ANPs show enhanced expressiveness relative to the NP and is able to model a wider range of functions.

## 2  BACKGROUND

### 2.1  NEURAL PROCESSES

The NP is a model for regression functions that map an input $x_i \in \mathbb{R}^{d_x}$ to an output $y_i \in \mathbb{R}^{d_y}$. In particular, the NP defines a (infinite) family of conditional distributions, where one may condition on an arbitrary number of observed *contexts* $(x_C, y_C) := (x_i, y_i)_{i \in C}$ to model an arbitrary number of *targets* $(x_T, y_T) := (x_i, y_i)_{i \in T}$ in a way that is invariant to ordering of the contexts and ordering of the targets. The model is defined for arbitrary $C$ and $T$ but in practice we use $C \subset T$. The deterministic NP models these conditional distributions as:

$$p(y_T | x_T, x_C, y_C) := p(y_T | x_T, r_C) \tag{1}$$

with $r_C := r(x_C, y_C) \in \mathbb{R}^d$ where $r$ is a deterministic function that aggregates $(x_C, y_C)$ into a finite dimensional representation with permutation invariance in $C$. In practice, each context $(x, y)$ pair is passed through an MLP to form a representation of each pair, and these are aggregated by taking the mean to form $r_C$. The likelihood $p(y_T | x_T, r_C)$ is modelled by a Gaussian factorised across the targets $(x_i, y_i)_{i \in T}$ with mean and variance given by passing $x_i$ and $r_C$ through an MLP. The unconditional distribution $p(y_T | x_T)$ (when $C = \varnothing$) is defined by letting $r_\varnothing$ be a fixed vector.

The latent variable version of the NP model includes a global latent $z$ to account for uncertainty in the predictions of $y_T$ for a given observed $(x_C, y_C)$. It is incorporated into the model via a *latent path* that complements the *deterministic path* described above. Here $z$ is modelled by a factorised Gaussian parametrised by $s_C := s(x_C, y_C)$, with $s$ being a function of the same properties as $r$

$$p(y_T | x_T, x_C, y_C) := \int p(y_T | x_T, r_C, z) q(z | s_C) dz \tag{2}$$

with $q(z | s_\varnothing) := p(z)$, the prior on $z$. The likelihood is referred to as the *decoder*, and $q, r, s$ form the *encoder*. See Figure 2 for diagrams of these models.

The motivation for having a global latent is to model different realisations of the data generating stochastic process — each sample of $z$ would correspond to one realisation of the stochastic process. One can define the model using either just the deterministic path, just the latent path, or both. In this

work we investigate the case of using both paths, which gives the most expressive model and also gives a sensible setup for incorporating attention, as we will show later in Section 3.

The parameters of the encoder and decoder are learned by maximising the following ELBO

$$\log p(\boldsymbol{y}_T | \boldsymbol{x}_T, \boldsymbol{x}_C, \boldsymbol{y}_C) \geq \mathbb{E}_{q(\boldsymbol{z}|\boldsymbol{s}_T)}[\log p(\boldsymbol{y}_T | \boldsymbol{x}_T, \boldsymbol{r}_C, \boldsymbol{z})] - D_{\mathrm{KL}}(q(\boldsymbol{z}|\boldsymbol{s}_T) \| q(\boldsymbol{z}|\boldsymbol{s}_C)) \qquad (3)$$

for a random subset of contexts $C$ and targets $T$ via the reparametrisation trick (Kingma & Welling, 2014; Rezende et al., 2014). In other words, the NP learns to reconstruct targets, regularised by a KL term that encourages the summary of the contexts to be not too far from the summary of the targets. This is sensible since we are assuming that the contexts and targets come from the same realisation of the data-generating stochastic process, and especially so if targets contain contexts. At each training iteration, the number of contexts and targets are also chosen randomly (as well as being randomly sampled from the training data), so that the NP can learn a wide family of conditional distributions.

NPs have many desirable properties, namely (i) **Scalability**: computation scales linearly at $O(n+m)$ for $n$ contexts and $m$ targets at train and prediction time. (ii) **Flexibility**: defines a very wide family of distributions, where one can condition on an arbitrary number of contexts to predict an arbitrary number of targets. (iii) **Permutation invariance**: the predictions of the targets are order invariant in the contexts. However these advantages come at the cost of not satisfying consistency in the contexts. For example, if $\boldsymbol{y}_{1:m}$ is generated given some context set, then its distribution need not match the distribution you would obtain if $\boldsymbol{y}_{1:n}$ is generated first, appended to the context set then $\boldsymbol{y}_{n+1:m}$ is generated. However maximum-likelihood learning can be interpreted as minimising the KL between the (consistent) conditional distributions of the data-generating stochastic process and the corresponding conditional distributions of the NP. Hence we could view the NP as approximating the conditionals of the consistent data-generating stochastic process.

## 2.2 ATTENTION

Given a set of key-value pairs $(k_i, v_i)_{i \in \mathcal{I}}$ and a query $q$, an attention mechanism computes weights of each key with respect to the query, and aggregates the values with these weights to form the value corresponding to the query. In other words, the query *attends* to the key-value pairs. The queried values are invariant to the ordering of the key-value pairs; this permutation invariance property of attention is key in its application to NPs. The idea of using a differentiable addressing mechanism that can be learned from the data has been applied successfully in various areas of Deep Learning, namely handwriting generation and recognition (Graves, 2012) and neural machine translation (Bahdanau et al., 2015). More recently, there has been work employing *self-attention* (where keys and queries are identical) to give expressive sequence-to-sequence mappings in natural language processing (Vaswani et al., 2017) and image modelling (Parmar et al., 2018).

We give some examples of attention mechanisms which are used in the paper. Suppose we have $n$ key-value pairs arranged as matrices $K \in \mathbb{R}^{n \times d_k}$, $V \in \mathbb{R}^{n \times d_v}$, and $m$ queries $Q \in \mathbb{R}^{m \times d_k}$. Simple forms of attention based on locality (weighting keys according to distance from query) are given by various stationary kernels. For example, the (normalised) *Laplace* kernel gives the queried values as

$$\mathbf{Laplace}(Q, K, V) := WV \in \mathbb{R}^{m \times d_v}, \qquad W_{i\cdot} := \mathrm{softmax}((-\|Q_{i\cdot} - K_{j\cdot}\|_1)_{j=1}^n) \in \mathbb{R}^n$$

Similarly (scaled) *dot-product* attention uses the dot-product between the query and keys as a measure of similarity, and weights the keys according to the values

$$\mathbf{DotProduct}(Q, K, V) := \mathrm{softmax}(QK^\top / \sqrt{d_k})V \in \mathbb{R}^{m \times d_v}$$

The use of dot-product attention allows the query values to be computed with two matrix multiplications and a softmax, allowing for use of highly optimised matrix multiplication code.

*multihead* attention (Vaswani et al., 2017) is a parametrised extension where for each head, the keys, values and queries are linearly transformed, then dot-product attention is applied to give head-specific values. These values are concatenated and linearly transformed to produce the final values:

$$\mathbf{MultiHead}(Q, K, V) := \mathrm{concat}(\mathrm{head}_1, \dots, \mathrm{head}_H)W \in \mathbb{R}^{m \times d_v}$$

$$\text{where } \mathrm{head}_h := \mathrm{DotProduct}(QW_h^Q, KW_h^K, VW_h^V) \in \mathbb{R}^{m \times d_v}$$

This multihead architecture allows the query to attend to different keys for each head and tends to give smoother query-values than dot-product attention (c.f. Section 4).

# 3 ATTENTIVE NEURAL PROCESSES

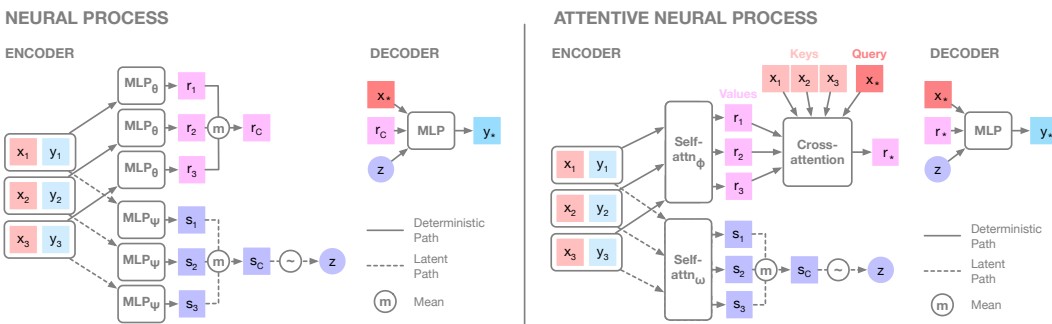

Figure 2: Model architecture for the NP (left) and Attentive NP (right)

Figure 2 describes how attention is incorporated into NP to give the Attentive NP (ANP). In summary, *self-attention* is applied to the context points to compute representations of each $(\boldsymbol{x}, \boldsymbol{y})$ pair, and the target input attends to these context representations (*cross-attention*) to predict the target output. In detail, the representation of each context pair $(\boldsymbol{x}_i, \boldsymbol{y}_i)_{i \in C}$ before the mean-aggregation step is computed by a self-attention mechanism, in both the deterministic and latent path. The intuition for the self-attention is to model interactions between the context points. For example, if many context points overlap, then the query need not attend to all of these points, but only give high weight to one or a few. The self-attention will help obtain richer representations of the context points that encode these types of relations between the context points. We model higher order interactions by simply stacking the self-attention, as is done in Vaswani et al. (2017).

In the deterministic path, the mean-aggregation of the context representations that produces $\boldsymbol{r}_C$ is replaced by a *cross-attention* mechanism, where each target query $\boldsymbol{x}_*$ attends to the context $\boldsymbol{x}_C :=$ $(\boldsymbol{x}_i)_{i \in C}$ to produce a query-specific representation $\boldsymbol{r}_* := r^*(\boldsymbol{x}_C, \boldsymbol{y}_C, \boldsymbol{x}_*)$. This is precisely where the model allows each query to attend more closely to the context points that it deems relevant for the prediction. The reason we do not have an analogous mechanism in the latent path is that we would like to preserve the global latent, that induces dependencies between the target predictions. The interpretation of the latent path is that $\boldsymbol{z}$ gives rise to correlations in the marginal distribution of the target predictions $\boldsymbol{y}_T$, modelling the global structure of the stochastic process realisation, whereas the deterministic path models the fine-grained local structure.

The decoder remains the same, except we replace the shared context representation $\boldsymbol{r}_C$ with the query-specific representation $\boldsymbol{r}_*$. Note that permutation invariance in the contexts is preserved with the attention mechanism. If we use uniform attention (all contexts given the same weight) throughout, we recover the NP. ANP is trained using the same loss (3) as the original NP, also using Gaussian likelihood $p(\boldsymbol{y}_i|\boldsymbol{x}_i, r^*(\boldsymbol{x}_C, \boldsymbol{y}_C, \boldsymbol{x}_i), \boldsymbol{z})$ and diagonal Gaussian $q(\boldsymbol{z}|\boldsymbol{s}_C)$.

The added expressivity and resulting accuracy of the NP with attention comes at a cost. The computational complexity is raised from $O(n + m)$ to $O(n(n + m))$, since we apply self-attention across the contexts and for every target point we compute weights for all contexts. However most of the computation for the (self-)attention is done via matrix multiplication (c.f. Section 2.2), and so can be done in parallel across the contexts and across the targets. In practice, the training time for ANPs remains comparable to NPs, and in fact we show that ANPs learn significantly faster than NPs not only in terms of training iterations but also in wall-clock time, despite being slower at prediction time (c.f. Section 4).

# 4 EXPERIMENTAL RESULTS

Note that the (A)NP learns a stochastic process, so should be trained on multiple functions that are realisations of the stochastic process. At each training iteration, we draw a batch of realisations from the data generating stochastic process, and select random points on these realisations to be the targets and a subset to be the contexts to optimise the loss in Equation (3). We use the same decoder architecture for all experiments, and 8 heads for multihead. See Appendix A for architectural details.

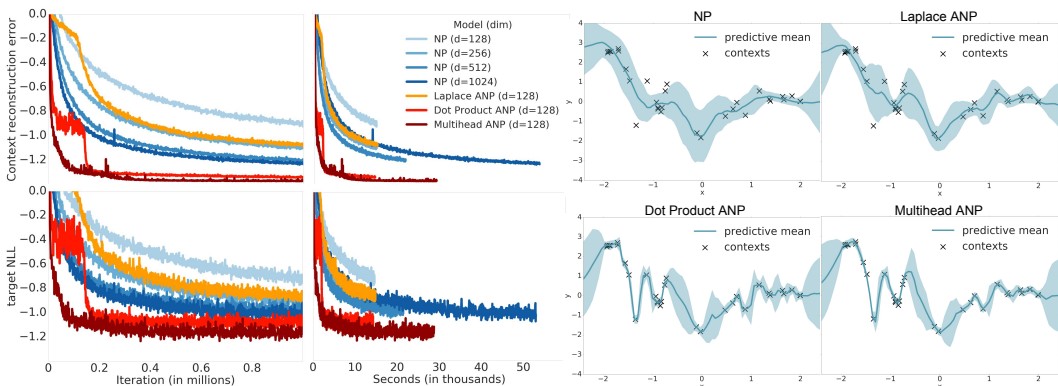

Figure 3: Qualitative and quantitative results of different attention mechanisms for 1D GP function regression with random kernel hyperparameters. **Left**: moving average of context reconstruction error (top) and target negative log likelihood (NLL) given contexts (bottom) plotted against training iterations (left) and wall clock time (right). $d$ denotes the bottleneck size i.e. hidden layer size of all MLPs and the dimensionality of $r$ and $z$. **Right**: predictive mean and variance of different attention mechanisms given the same context. Best viewed in colour.

**1D Function regression on synthetic GP data** We first explore the (A)NPs trained on data that is generated from a Gaussian Process with a squared-exponential kernel and small likelihood noise[1]. We emphasise that (A)NPs need not be trained on GP data or data generated from a known stochastic process, and this is just an illustrative example. We explore two settings: one where the hyperparameters of the kernel are fixed throughout training, and another where they vary randomly at each training iteration. The number of contexts ($n$) and number of targets ($m$) are chosen randomly at each iteration ($n \sim U[3, 100]$, $m \sim n + U[0, 100 - n]$). Each $x$-value is drawn uniformly at random in $[-2, 2]$. For this simple 1D data, we do not use self-attention and just explore the use of cross-attention in the deterministic path (c.f. Figure 2). Thus we use the same encoder/decoder architecture for NP and ANP, except for the cross-attention. See Appendix B for experimental details.

Figure 3 (left) shows context reconstruction error $\frac{1}{|C|} \sum_{i \in C} \mathbb{E}_{q(\boldsymbol{z}|\boldsymbol{s}_C)}[\log p(\boldsymbol{y}_i|\boldsymbol{x}_i, r^*(\boldsymbol{x}_C, \boldsymbol{y}_C, \boldsymbol{x}_i), \boldsymbol{z})]$ and NLL of targets given contexts $\frac{1}{|T|} \sum_{i \in T} \mathbb{E}_{q(\boldsymbol{z}|\boldsymbol{s}_C)}[\log p(\boldsymbol{y}_i|\boldsymbol{x}_i, r^*(\boldsymbol{x}_C, \boldsymbol{y}_C, \boldsymbol{x}_i), \boldsymbol{z})]$ for the different attention mechanisms, trained on a GP with random kernel hyperparameters. ANP shows a much more rapid decrease in reconstruction error and lower values at convergence compared to the NP, especially for dot product and multihead attention. This holds not only against training iteration but also against wall clock time, so learning is fast despite the added computational cost of attention. The right column plots show that the computation times of Laplace and dot-product ANP are similar to the NP for the same value of $d$, and multihead ANP takes around twice the time. We also show how the size of the bottleneck ($d$) in the deterministic and latent paths of the NP affects the underfitting behaviour of NPs. The figure shows that raising $d$ does help achieve better reconstructions, but there appears to be a limit in how much reconstructions can improve. Beyond a certain value of $d$, the learning for the NP becomes too slow, and the value of reconstruction error at convergence is still higher than that achieved by multihead ANP with 10% of the wall-clock time. Hence using ANPs has significant benefits over simply raising the bottleneck size in NPs.

In Figure 3 (right) we visualise the learned conditional distribution for a qualitative comparison of the attention mechanisms. The context is drawn from the GP with the hyperparameter values that give the most fluctuation. Note that the predictive mean of the NP underfits the context, and tries to explain the data by learning a large likelihood noise. Laplace shows similar behaviour, whereas dot-product attention gives predictive means that accurately predict almost all context points. Note that Laplace attention is parameter-free (keys and queries are the x-coordinates) whereas for dot-product attention we have set the keys and queries to be parameterised representations of the x-values (output of learned MLP that takes x-coordinates as inputs). So the dot-product similarities are computed in a learned representation space, whereas for Laplace attention the similarities are computed based on L1 distance in the x-coordinate domain, hence it is expected that dot-product attention outperforms Laplace attention. However dot-product attention displays non-smooth predictions, shown more

---

[1]Code is available at `https://github.com/deepmind/neural-processes/blob/master/attentive_neural_process.ipynb`

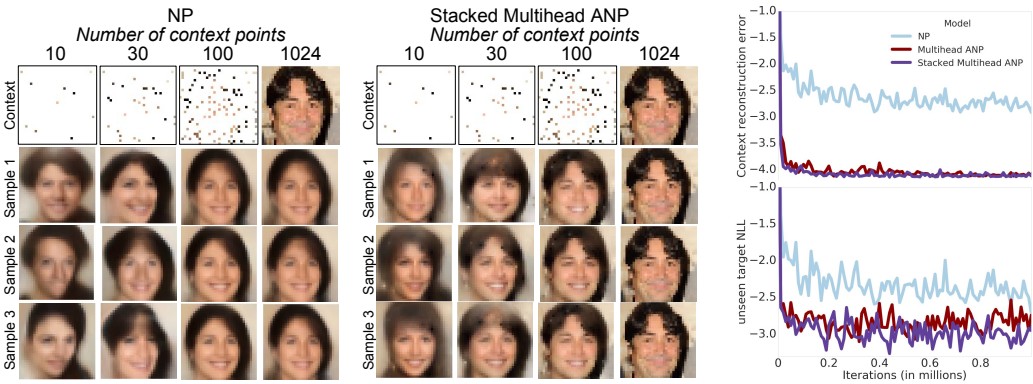

(a) Reconstructions of full CelebA image from a varying number of random context points for NP (left) and *Stacked Multihead* ANP (right).

(b) Context NLL (top) and unseen target NLL given contexts (bottom).

Figure 4: Qualitative and quantitative results on test set for 2D CelebA function regression.

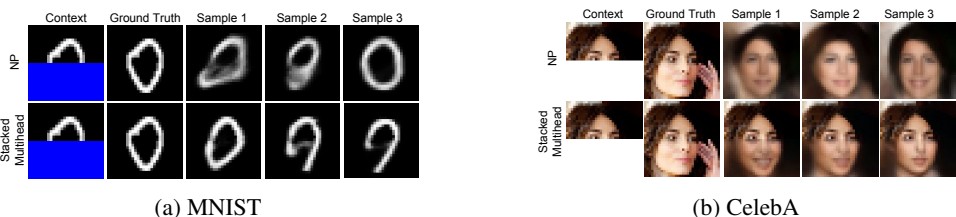

(a) MNIST

(b) CelebA

Figure 5: Reconstruction of full image from top half. The CelebA results use the **same models** (with the **same parameter values**) as Figure 4a.

clearly in the predictive standard deviations (c.f. Appendix C for an explanation). The multiple heads in multihead attention appear to help smooth out the interpolations, giving good reconstruction of the contexts as well as prediction of the targets, while preserving increased predictive uncertainty away from the contexts as in a GP. The results for (A)NP trained on fixed GP kernel hyperparameters are similar (c.f. Appendix C), except that the NP underfits to a lesser degree because of the reduced variety of sample curves (functions) in the data. This difference in performance for the two kernel hyperparameter settings provides evidence of how the ANP is more expressive than the NP and can learn a wider range of functions.

Using the trained (A)NPs we tackle a toy Bayesian Optimisation (BO) problem, where the task is to find the minimum of test functions drawn from a GP prior. This is a proof-of-concept experiment showing the utility of being able to sample entire functions from the (A)NP and having accurate context reconstructions. See Appendix C for the details and an analysis of results.

**2D Function regression on image data** Image data can also be interpreted as being generated from a stochastic process (since there are dependencies between pixel values), and predicting the pixel values can be cast as a regression problem mapping a 2D pixel location $\boldsymbol{x}_i$ to its pixel intensity $\boldsymbol{y}_i$ ($\in \mathbb{R}^1$ for greyscale, $\in \mathbb{R}^3$ for RGB). Each image corresponds to one realisation of the process sampled on a fixed 2 dimensional grid. We train the ANP on MNIST (LeCun et al., 1998) and $32 \times 32$ CelebA (Liu et al., 2015) using the standard train/test split with up to 200 context/target points at training. For this application we explore the use of self-attentional layers in the encoder, stacking them as is done in Parmar et al. (2018). See Appendix D for experimental details.

On both datasets we show results of three different models: NP, ANP with multihead cross-attention in the deterministic path (*Multihead* ANP), and ANP with both multihead attention in the deterministic path and two layers of stacked self-attention in both the deterministic and latent paths (*Stacked Multihead* ANP). Figure 4a shows predictions of the full image (i.e. full target) with a varying number of random context pixels, from 10 to 1024 (full image) for a randomly selected image (see Appendix E for other images). For each we generate predictions that correspond to the mean of $p(\boldsymbol{y}_T|\boldsymbol{x}_T, \boldsymbol{r}_C, \boldsymbol{z})$ for three different samples of $\boldsymbol{z} \sim q(\boldsymbol{z}|\boldsymbol{s}_C)$. The NP (left) gives reasonable predictions with a fair amount of diversity for fewer contexts, but the reconstructions of the whole image are not accurate, compared to *Stacked Multihead* ANP (right) where the reconstructions are indistinguishable from the original. The use of attention also helps achieve crisper inpaintings when the

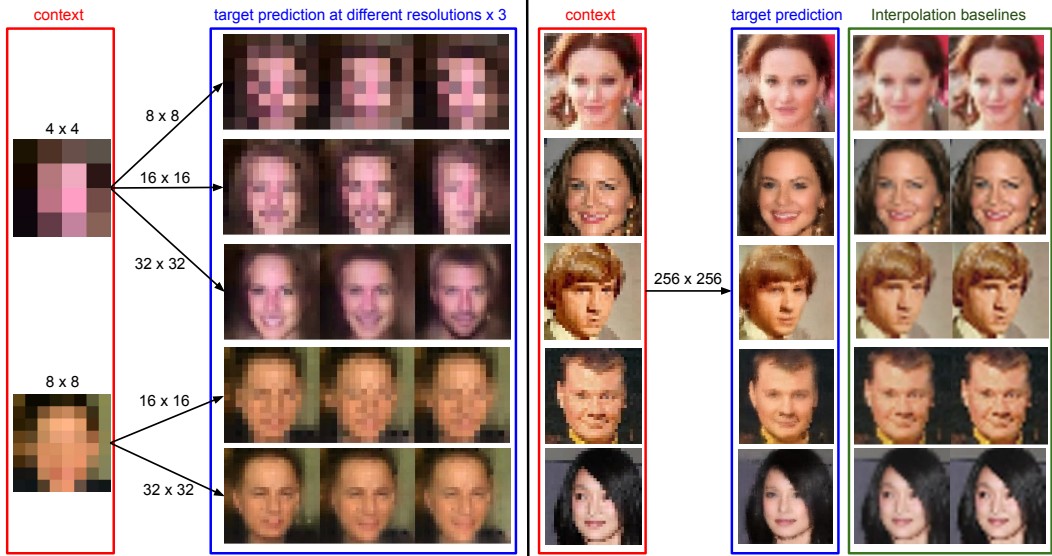

Figure 6: Mapping between different resolutions by the **same model** (with the **same parameter values**) as *Stacked Multihead* ANP in Figures 4a, 5b. The two rightmost columns show the results of baseline methods, namely linear and cubic interpolation to $256 \times 256$.

target pixels are filled in, enhancing the ANP's ability to model less smooth 2D functions compared to the NP. The diversity in faces and digits obtained with different values of $z$ is apparent the different samples, providing evidence for the claim that $z$ can model global structure of the image, with one sample corresponding to one realisation of the data generating stochastic process. Similar conclusions hold for MNIST (see Appendix E) and for the full image prediction using the top half as context in Figure 5. In the latter task, note that the model has never been trained on more than 200 context points, yet it manages to generalise to when the context is of size 512 (half the image).

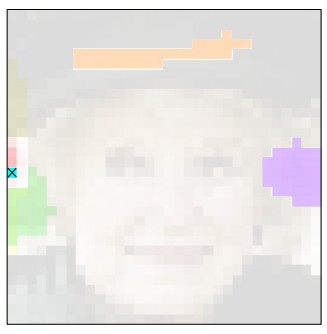

Figure 7: Pixels attended to by each head of multi-head attention in *Multihead* ANP given a target pixel. Each head is given a different colour and the target pixel is marked with a cross.

Figure 4b verifies quantitatively that both *Multihead* and *Stacked Multihead* ANP give a much improved context reconstruction error compared to the NP. Similarly the NLL for the target points (that are not included in the context) is improved with multihead cross-attention, showing small gains with stacked self-attention. However qualitatively, there are noticeable gains in crispness and global coherence when using stacked self-attention (see Appendix E). In Figure 7 we visualise each head of *Multihead* ANP for CelebA. We let the target pixel (cross) attend to all pixels, and see where each head of the attention focuses on. We colour-code the pixels with the top 20 weights per head, with intensity proportional to the attention weight. We can see that each head has different roles: the cyan head only looks at the target pixel and nothing else; the red head looks at a few pixels nearby; the green head looks at a larger region nearby; the yellow looks at the pixels on the column of the target; the orange looks at some band of the image; the purple head (interestingly) looks at the other side of the image, trying to exploit the symmetry of faces. We observe consistent behaviour in these heads for other target pixels (see Figure 16 of Appendix E).

One other illustrative application of (A)NPs trained on images is that one can map images from one resolution to another, even if the model has only been trained on one resolution. Because the two dimensional $x$ (pixel locations) are modelled as real values that live in a continuous space, the model can predict the $y$ (pixel intensities) of any point in this space, and not just the grid of points that it was trained on. Hence using one grid as the context and a finer grid as the target, the model can map a given resolution to a higher resolution. This could, however, be problematic for NPs whose reconstructions can be inaccurate, so the prediction of the target resolution can look very different to the original image (see Figure 19 of Appendix E). The reconstructions of ANPs may be accurate enough to give reliable mappings

between different resolutions. We show results for such mappings given by the same *Stacked Multihead* ANP (the same model used to produce Figures 4a, 5b) in Figure 6. On the left, we see that the ANP (trained on $32 \times 32$ images) is capable of mapping low resolutions ($4 \times 4$ or $8 \times 8$) to fairly realistic $32 \times 32$ target outputs with some diversity for different values of $z$ (more diversity for the $4 \times 4$ contexts as expected). Perhaps this performance is to be expected since the model has been trained on data that has $32 \times 32$ resolution. The same model allows us to map to even higher resolutions, namely from the original $32 \times 32$ images to $256 \times 256$, displayed on the right of the figure. We see that even though the model has never seen any images beyond the original resolution, the model learns a fairly realistic high resolution image with sharper edges compared to the baseline interpolation methods. Moreover, there is some evidence that it learns an internal representation of the appearance of faces, when for example it learns to fill in the eye even when the original image is too coarse to separate the iris (coloured part) from the sclera (white part) (e.g. top row image), a feature that is not possible with simple interpolation. See Figure 19 in Appendix E for larger versions of the images.

For each of MNIST and CelebA, all qualitative plots in this section were given from the **same model** (with the **same parameter values**) for each attention mechanism, learned by optimising the loss in Equation (3) over random context pixels and random target pixels at each iteration. It is important to note that we do not claim the ANP to be a replacement of state of the art algorithms of image inpainting or super-resolution, and rather we show these image applications to highlight the flexibility of the ANP in modelling a wide family of conditional distributions.

## 5    RELATED WORK

The work related to NPs in the domain of Gaussian Processes, Meta-Learning, conditional latent variable models and Bayesian Learning have been discussed extensively in the original works of Garnelo et al. (2018a;b), hence we focus on works that are particularly relevant for ANPs.

**Gaussian Processes (GPs)** Returning to our motivation for using attention in NPs, there is a clear parallel between GP kernels and attention, in that they both give a measure of similarity between two points in the same domain. The use of attention in an embedding space that we explore is related to Deep Kernel Learning (Wilson et al., 2016) where a GP is applied to learned representations of data. Here, however, learning is still done in a GP framework by maximising the marginal likelihood. We reiterate that the training regimes of GPs and NPs are different, so a direct comparison between the methods is difficult. One possibility for comparison is to learn the GP via the training regime of NPs, namely updating the kernel hyperparameters at each iteration via one gradient step of the marginal likelihood on the mini-batch of data. However, this would still have a $O(n^3)$ computational cost in the naive setting and may require kernel approximations. In general, the predictive uncertainties of GPs depend heavily on the choice of the kernel, whereas NPs learn predictive uncertainties directly from the data. Despite these drawbacks, GPs have the benefit of being consistent stochastic processes, and the covariance between the predictions at different $x$-values and the marginal variance of each prediction can be expressed exactly in closed form, a feature that the current formulation of (A)NPs do not have. Variational Implicit Processes (VIP) (Ma et al., 2018) are also related to NPs, where VIP defines a stochastic process using the same decoder setup with a finite dimensional $z$. Here, however, the process and its posterior given observed data are both approximated by a GP and learned via a generalisation of the Wake-Sleep algorithm (Hinton et al., 1995).

**Meta-Learning** (A)NPs can be seen as models that do few-shot learning, although this is not the focus of our work. Given input-output pairs drawn from a new function at test time, one can reason about this function by looking at the predictive distribution conditioning on these input-output pairs. There is a plethora of works in few-shot classification, of which Vinyals et al. (2016); Snell et al. (2017); Santoro et al. (2016) use attention to locate the relevant observed image/prototype given a query image. Attention has also been used for tasks in Meta-RL such as continuous control and visual navigation (Mishra et al., 2018). Few-shot density estimation using attention has also been explored extensively in numerous works (Rezende et al., 2016; Reed et al., 2017; Bornschein et al., 2017; Bartunov & Vetrov, 2018). Especially relevant are the Neural Statistician (Edwards & Storkey, 2017) and the Variational Homoencoder (Hewitt et al., 2018) who have a similar permutation invariant encoder (that outputs summaries of a data set), but use local latents on top of a global latent. For ANPs, we look at the less-explored regression setting. The authors of Vfunc (Bachman

et al., 2018) also explore regression on a toy 1D domain, using a similar setup to NPs but optimising an approximation to the entropy of the latent function, without any attention mechanisms. Multi-task learning has also been tackled in the GP literature by various works (Teh et al., 2005; Bonilla et al., 2008; Alvarez et al., 2012; Dai et al., 2017).

**Generative Query Networks** (Eslami et al., 2018; Kumar et al., 2018) are models for spatial prediction that render a frame of a scene given a viewpoint. Their model corresponds to a special case of NPs where the $x$ are viewpoints and the $y$ are frames of a scene. Rosenbaum et al. (2018) apply the GQN to the task of 3D localisation with an attention mechanism, but attention is applied to patches of context frames ($y$) instead of a parametric representation of viewpoints ($x$). Note that in our work the targets attend to the contexts via the $x$.

## 6 CONCLUSION AND DISCUSSION

We have proposed ANPs, which augment NPs with attention to resolve the fundamental problem of underfitting. We have shown that this greatly improves the accuracy of predictions in terms of context and target NLL, results in faster training, and expands the range of functions that can be modelled. There is a wide scope of future work for ANPs. Regarding model architecture, one way of incorporating cross-attention into the latent path and modelling the dependencies across the resulting local latents is to also have a global latent, much like the setup of the Neural Statistician but translated to the regression setting. An interesting further application would be to train ANPs on text data, enabling them to fill in the blanks in a stochastic manner. For the image application, the Image Transformer (ImT) (Parmar et al., 2018) has some interesting connections with ANPs: its local self-attention used to predict consecutive pixel blocks from previous blocks has parallels with how our model attends to context pixels to predict target pixels. Replacing the MLP in the decoder of the ANP with self-attention across the target pixels, we have a model that closely resembles an ImT defined on arbitrary orderings of pixels. This is in contrast to the original ImT, which presumes a fixed ordering and is trained autoregressively. We plan to equip ANPs with self-attention in the decoder, and see how far their expressiveness can be extended. In this setup, however, the targets will affect each other's predictions, so the ordering and grouping of the targets will become important.

ACKNOWLEDGMENTS

We would like to thank Ali Razavi for his advice on implementing multihead attention, and Michael Figurnov for helpful discussion.

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

## APPENDIX

## A    ARCHITECTURAL DETAILS FOR (A)NP

We show the architectural details of the NP and the Multihead ANP models used for the 1D and 2D regression experiments below in Figure 8. All MLPs have relu non-linearities except the final layer, which has no non-linearity. The latent path outputs $\mu_z, \omega_z \in \mathbb{R}^d$, which parameterises $q(\boldsymbol{z}|\boldsymbol{s}_C) = \mathcal{N}(\boldsymbol{z}|\mu_z, 0.1 + 0.9\sigma(\omega_z))$ where $\sigma$ is the sigmoid function. Similarly the decoder outputs $\mu_y, \omega_y$, which parameterises $p(\boldsymbol{y}_i|\boldsymbol{z}, \boldsymbol{x}_C, \boldsymbol{y}_C, \boldsymbol{x}_i) = \mathcal{N}(\boldsymbol{y}_i|\mu_y, 0.1 + 0.9f(\omega_y))$ where $f$ is the softplus function.

The 1D regression experiments use the basic formulation of multihead cross-attention (denoted $Multihead_1$) in Figure 8, whereas the 2D regression experiments uses a form of multihead cross-attention used in the Image Transformer (Parmar et al., 2018). The only difference is that we do not use dropout, to limit the stochasticity of the model to the latent $z$.

Self-attention uses the same architecture as cross-attention but with $k_i = v_i$, $q = k_j$ for each $j \in C$, to output $|C|$ representations given $|C|$ input representations. Since the self-attention module has the same number of inputs and outputs, it can be stacked. We stack 2 layers of self-attention for *Stacked Multihead ANP* in the 2D Image regression experiments. Stacking more layers did not lead to noticeable gains qualitatively and quantitatively.

## B    EXPERIMENTAL DETAILS OF 1D FUNCTION REGRESSION EXPERIMENT

For the squared exponential kernel of the data generating GP, we use a length scale $l = 0.6$ and kernel scale $\sigma_f^2 = 1$ for the fixed kernel hyperparameter experiments. For the random kernel hyperparameter case, we sample $l \sim U[0.1, 0.6]$, $\sigma_f \sim U[0.1, 1]$. For both, the likelihood noise is $\sigma_n = 0.02$. We use a batch size of 16 — in the fixed hyperparameter setting, we draw 16 curves from a GP with these hyperparameters, and in the random hyperparameter setting, we sample 16 random values of hyperparameters and draw a curve from GPs with each of these hyperparameters. We use the Adam Optimiser (Kingma & Ba, 2015) with a fixed learning rate of 5e-5 and Tensorflow defaults for the other hyperparameters. We use one sample of $q(z|\boldsymbol{s}_C)$ to form a MC estimate of the loss in Equation (3) during training and evaluation.

For NP, $d$ is varied between $\{128, 256, 512, 1024\}$ whereas for ANP we always use $d = 128$.

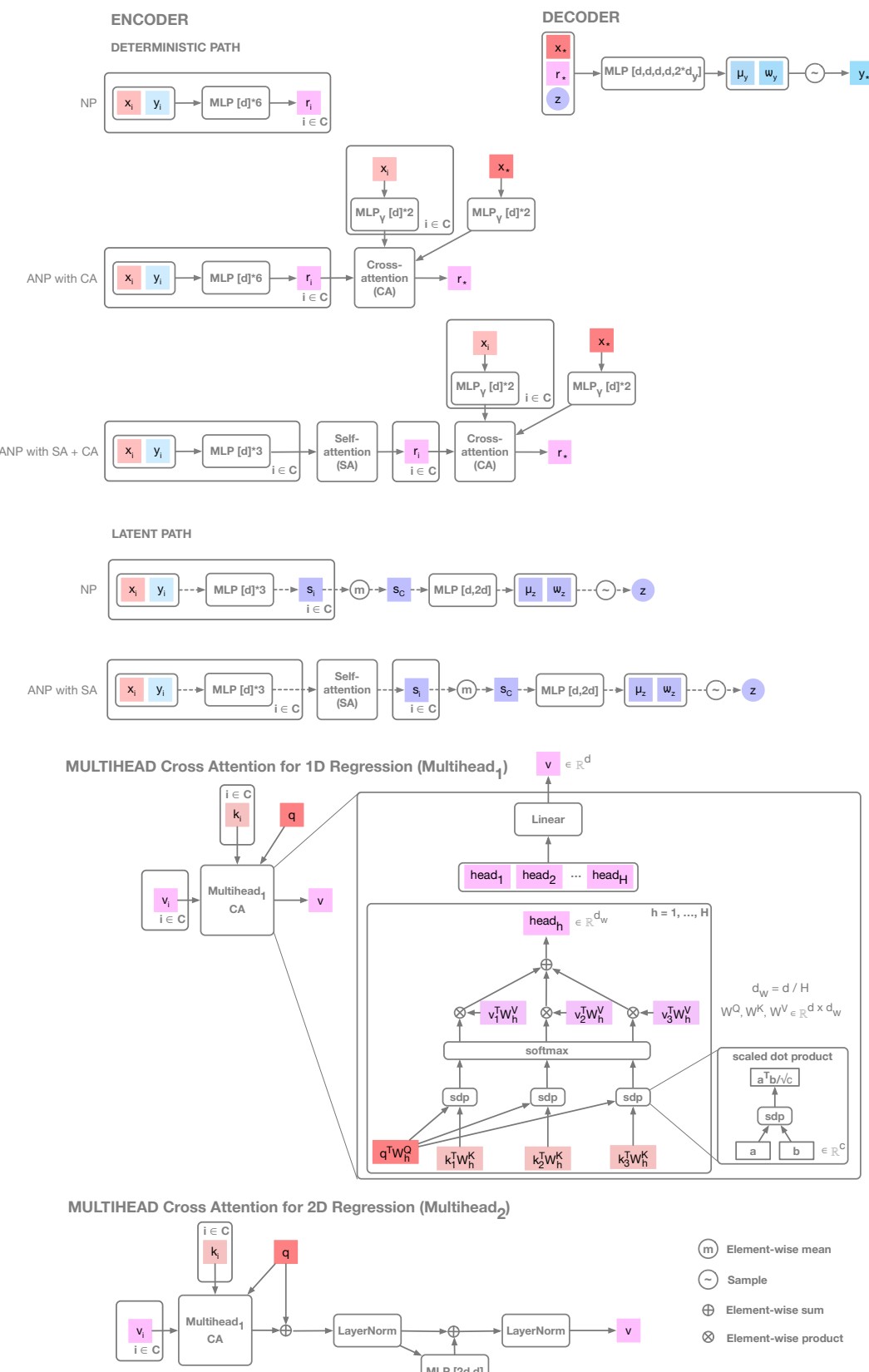

Figure 8: The model architecture for NP and ANP for both 1D and 2D regression.

## C    ADDITIONAL FIGURES FOR 1D REGRESSION ON GP DATA

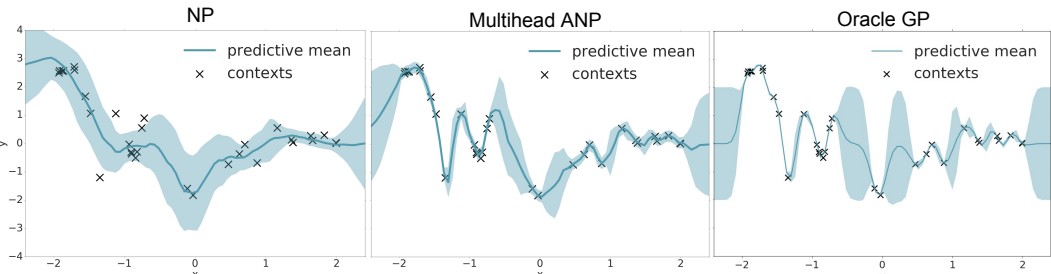

Figure 9: Same as right of Figure 3 but also comparing against the oracle GP from which context was drawn.

In Figure 9 we also compare the trained (A)NP models against the oracle GP from which the contexts were drawn. We see that the predictions Multihead ANP is notably closer to that of the oracle GP than the NP, but still underestimates the predictive variance. One possible explanation for this is that variational inference (used for learning the ANP) usually leads to underestimates of predictive variance. It would be interesting to investigate how this issue can be addressed.

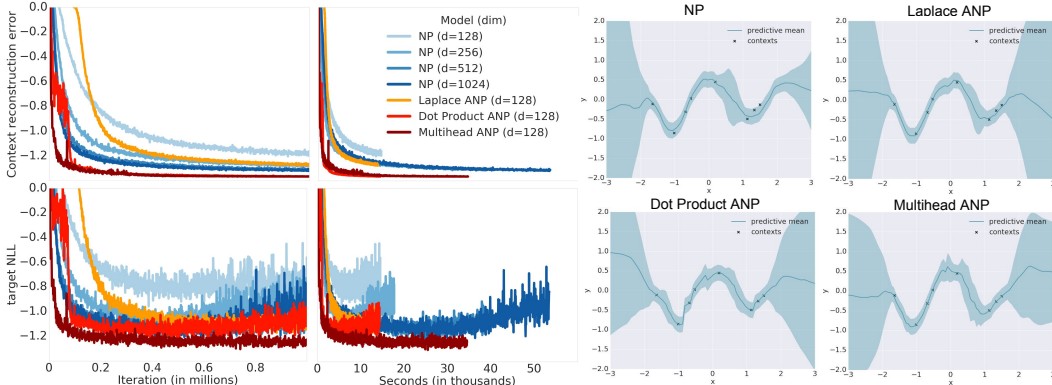

Figure 10: Same as Figure 3 but for fixed kernel hyperparameters.

The right of Figure 10 shows the conditional distributions for fixed kernel hyperparameters (with contexts drawn from the GP with these kernel hyperparameters), with highly non-smooth behaviour for dot-product attention as with the random kernel hyperparameter case. This behaviour seems to arise when the dot-product attention collapses to the local minimum of learning to be a nearest neighbour predictor (with one entry of the softmax becoming saturated), hence giving good reconstructions but poor interpolations between context points.

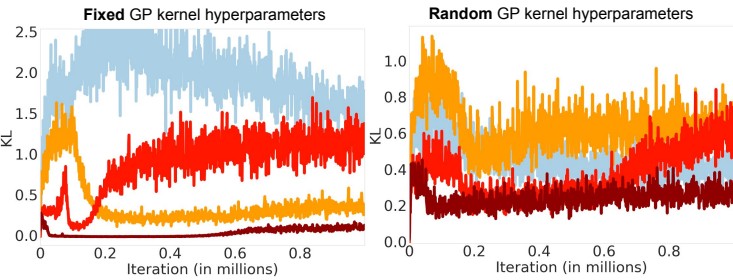

Figure 11: KL term in NP loss throughout training for data generated from a GP with fixed (left) and random (right) kernel hyperparameters, using the same colour scheme as Figure 10.

Figure 11 shows how the KL term in the (A)NP loss differs between training on the fixed kernel hyperparameter GP data and on the random kernel hyperparameter GP data. In the fixed hyperparameter case, the KL for multihead ANP quickly goes to 0, indicating that the model deems the deterministic path sufficient to make accurate predictions. However in the random hyperparameter case, there is added variation in the data, hence the attention gives a non-zero KL and uses the latents to model the uncertainty in the realisation of the stochastic process given some context points. In other words, given a context set, the model believes that there are multiple realisations of the stochastic process that can explain these contexts well, hence uses the latents to model this variation.

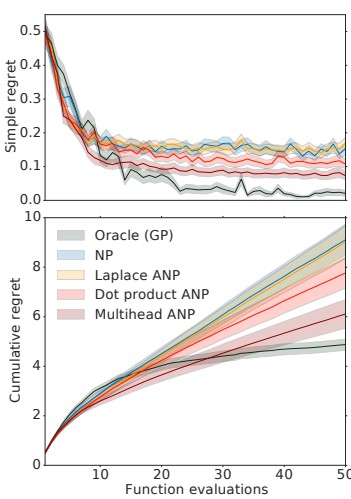

Figure 12: Simple and cumulative regret for BO.

Using the same (A)NPs trained on the 1D GP data, we tackle the BO problem of finding the minimum of test functions drawn from a GP prior. We compare ANPs trained with different attention mechanisms to an oracle GP for which we set the kernel hyperparameters to their true value. (A)NPs can be used for BO by considering all previous function evaluations as context points, thus obtaining an informed surrogate of the target function. While other choices are possible, we use Thompson sampling to drawing a simple function from the surrogate and acting according to its minimal predicted value. We show results averaged over 100 test functions in Figure 12. We can see that the simple regret (the difference between the predicted and true minimum) is consistently smallest for a NP with multihead attention, approaching the oracle GP. Among the NPs, the slope of the cumulative regret (simple regret summed up to given iteration) decreases most rapidly for multihead, indicating that previous function evaluations are being put to good use for subsequent predictions of the function minimum. The reason that the cumulative regret is initially lower than the oracle GP is a consequence of under-exploration, due to the uncertainties of ANP away from the context being smaller than that of the oracle GP.

## D EXPERIMENTAL DETAILS OF 2D IMAGE REGRESSION EXPERIMENT

Analogous to the 1D experiments, we take random pixels of a given image at training as targets, and select a subset of this as contexts, again choosing the number of contexts and targets randomly ($n \sim U[3, 200]$, $m \sim n + U[0, 200 - n]$). The $x$ are rescaled to $[-1, 1]$ and the $y$ are rescaled to $[-0.5, 0.5]$. We use a batch size of 16 for both MNIST and CelebA, i.e. use 16 randomly selected images for each batch. We use a learning rate of 5e-5 and 4e-5 respectively for MNIST and CelebA using the Adam optimiser with Tensorflow defaults for the other hyperparameters. The stacked self-attention architecture is the same as in the Image Transformer (Parmar et al., 2018), except that we do not use Dropout to restrict the stochasticity of the model to the global latent $z$, and do not use positional embeddings of the pixels. We use the same architecture for both Mnist and CelebA, and highlight that little tuning has been done regarding the architectural hyperparameters. We again use one sample of $q(z|s_C)$ to form a MC estimate of the loss in Equation (3) during training and evaluation.

## E ADDITIONAL FIGURES FOR 2D IMAGE REGRESSION ON MNIST AND CELEBA

We can see visually that the NP overestimates the predictive variance by looking at the plot of the standard deviation (bottom row) of Figure 13a. We see that the original NP shows noticeable uncertainty around the edges of the reconstruction for all context sets, whereas for the NP with attention, the uncertainty is reduced significantly as you increase the number of contexts until it almost disappears for the full context.

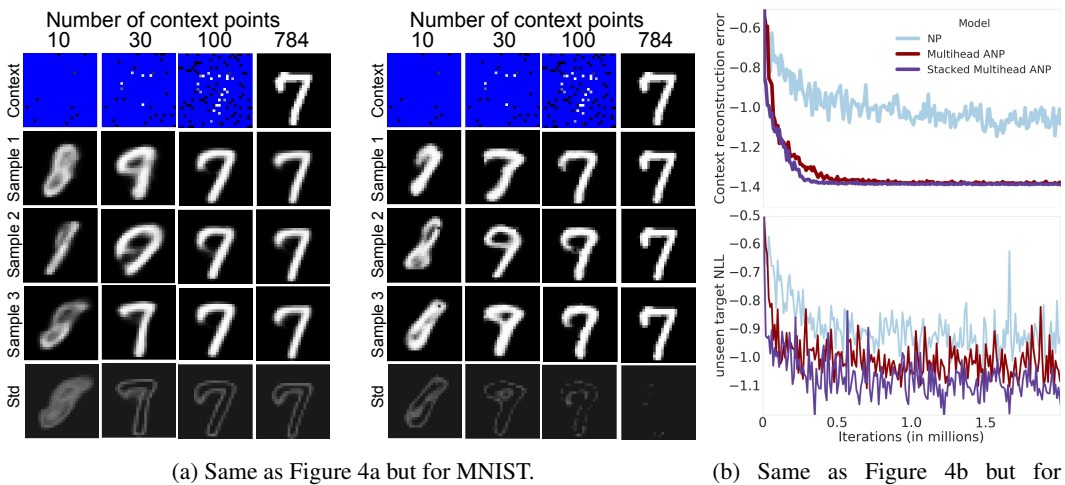

(a) Same as Figure 4a but for MNIST.

(b) Same as Figure 4b but for MNIST.

Figure 13: Qualitative and quantitative results of different attention mechanisms on test set for 2D MNIST function regression.

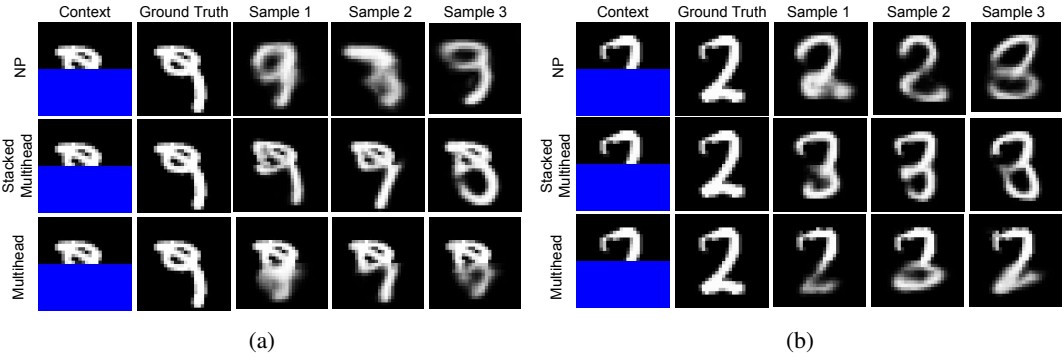

(a)                                                            (b)

Figure 14: More MNIST reconstruction of full image from top half.

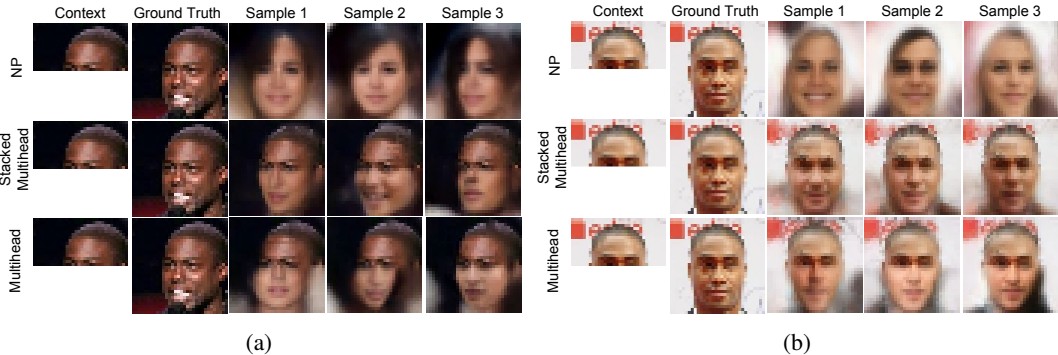

(a)                                                            (b)

Figure 15: More CelebA reconstruction of full image from top half.

From Figures 14 and 15 we see that *Stacked Multihead* ANP improves results significantly over *Multihead* ANP, giving sharper images with better global coherence even in the case where the face isn't axis-aligned (see Figure 15a).

Note that in Figure 7, the contexts contain the target, relying on the cyan head would be enough to give an accurate prediction, but the different roles of these heads also hold in the case where the target is disjoint from the context. This is shown in Figure 16 where the context is disjoint from the target. Here all heads become useful for the target prediction.

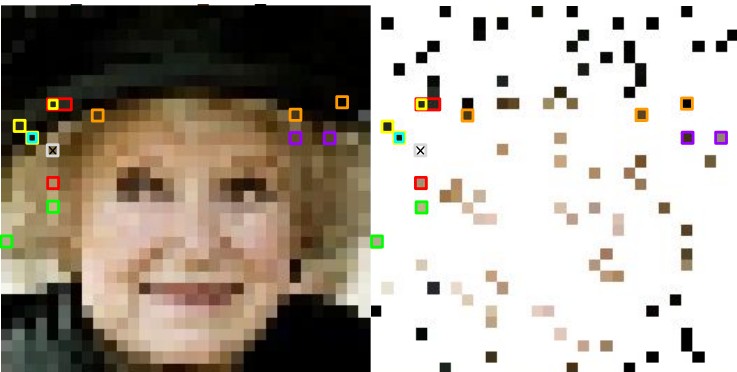

Figure 16: Visualisation of pixels attended by each head of multihead attention in the NP given a target pixel and a separate context of 100 random pixels. Each head is given a different colour (consistent with the colours in Figure 7 and the target pixel is marked by a cross.

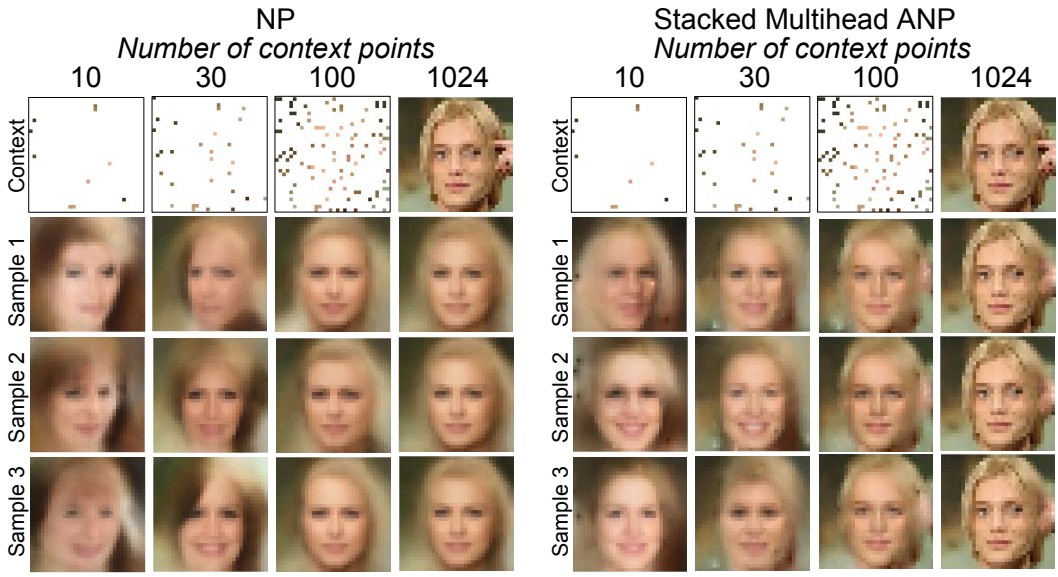

Figure 17: Same as Figure 4a but for a different image.

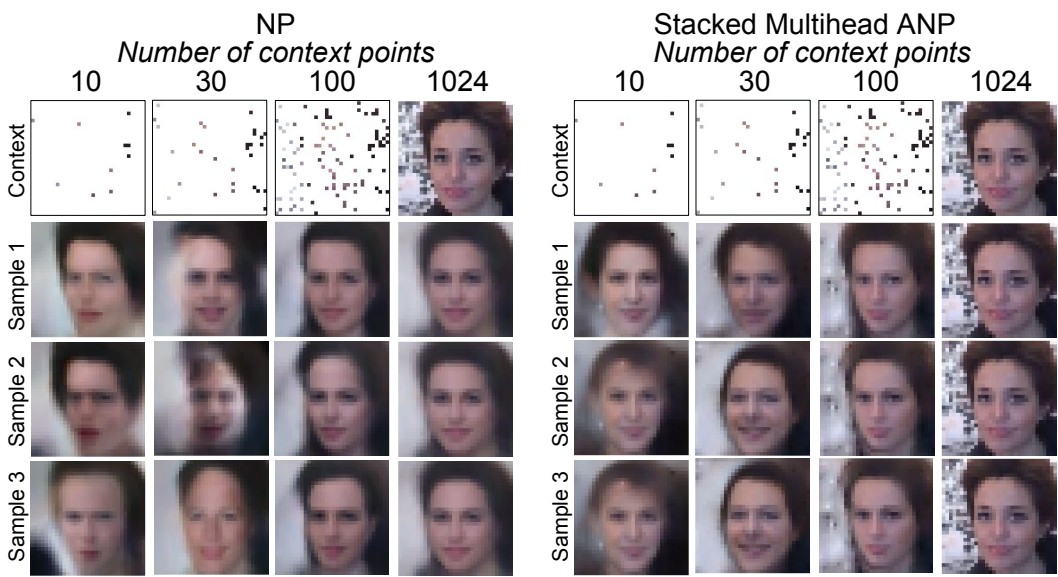

Figure 18: Same as Figure 4a but for a different image.

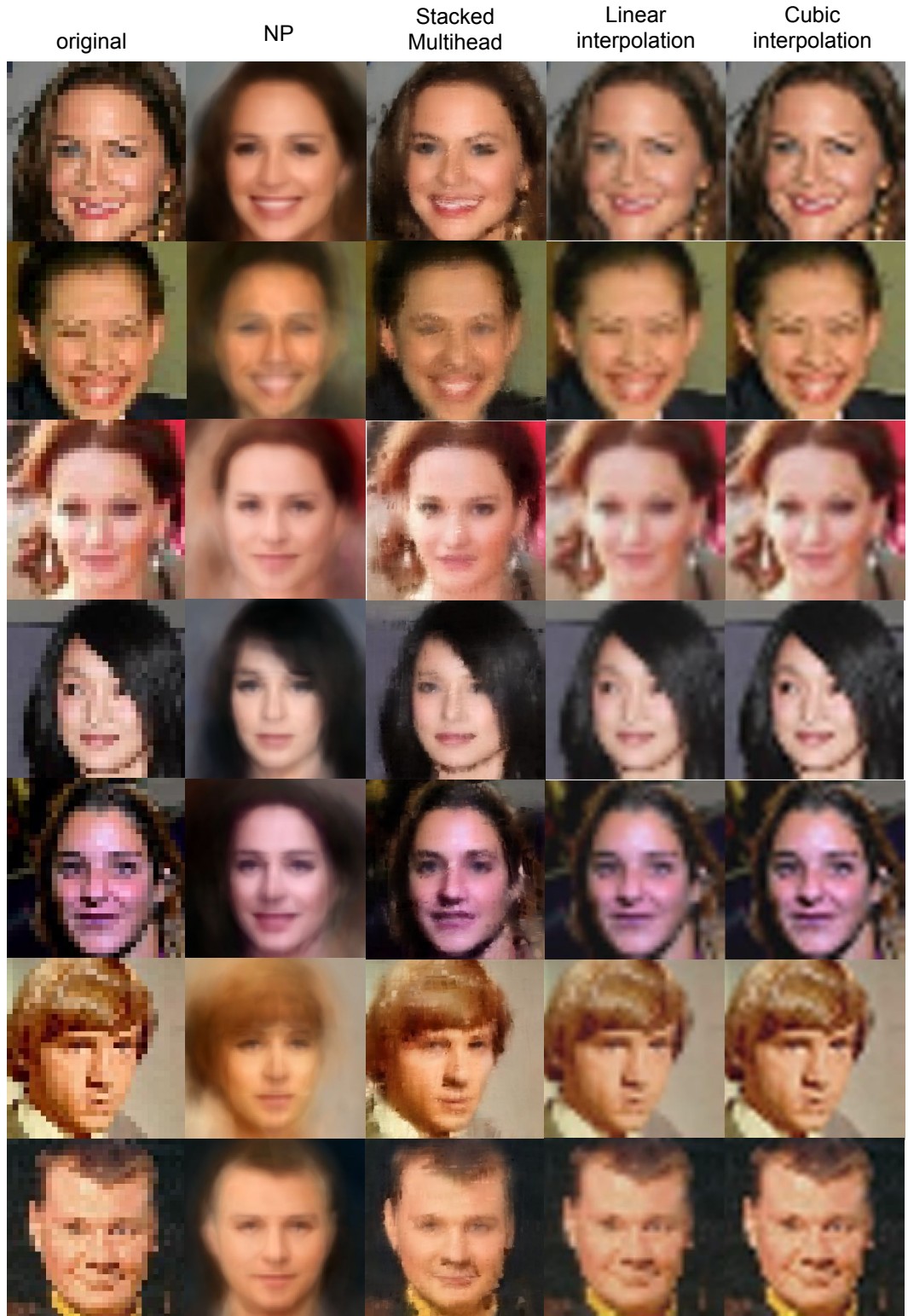

Figure 19: Mapping from $32 \times 32$ to $256 \times 256$ for different images.

