# OpenReview forum: "Attentive Neural Processes"
_ICLR.cc/2019/Conference_

### Official Review · AnonReviewer1 · 2018-11-02

**Rating:** 7
**Confidence:** 4

**Review:**

Summary:
The authors extend neural processes by incorporating two types of attention processes: self-attention for enriching the features of the context points and cross-attention for producing a query-specific representation. By replacing MLPs and mean pooling with these attention processes, the authors resolve the underfitting problem of NPs. The experimental results show that ANPs converge better and faster than NPs.

Overall, I had fun to read the paper and have not much to complain. Below are some comments and questions.

1. It is intuitive and reasonable that the cross-attention process makes ANPs fit with smaller predictive uncertainty for those regions with many context points. This is well illustrated in the qualitative results in the experiment section.

2. I would like to see an ablation study with the two separate techniques (self- and cross-attention processes) on NPs since the two techniques aim to improve different aspects of NPs. More specifically, I wonder the results of just adding cross-attention with the vanilla MLPs for feature encoding and just replacing the MLPs with self-attention modules while keeping using mean pooling.

3. While the dot product improves the performance significantly, the gain of Laplace is much lower. Also, qualitatively it fails to overcome the underfitting problem. Do you have any intuition about why performs worse than other models?

4. Do you have any specific application in mind? I just wonder some example tasks where contexts are given as inputs.

---

> ### Author Response · Authors · 2018-11-14
> **Response to AnonReviewer1**
>
> We would like to thank you for your thoughtful review with constructive criticism. Here are our responses to your suggestions and questions.
>
> “I would like to see an ablation study with the two separate techniques (self- and cross-attention processes) on NPs”
>
> Regarding the ablation study for using just cross-attention, all the ANP results for 1D GP regression use cross-attention but not self-attention (i.e. have MLP encodings of each context pair). Hence the improvements here show the contribution of cross-attention only. Figure 4b also shows quantitative results for the CelebA image regression with only cross-attention, denoted Multihead ANP in the legend. The counterpart results for MNIST can be found in Figure 13b of Appendix E of the revised version of the paper. Qualitative comparisons of the NP, Multihead ANP and Stacked Multihead ANP can be found in Figures 14 and 15. It would indeed be helpful to show results for the case when we just have self-attention in the encoder without any cross-attention. We are currently running these experiments and will include the results in the next version of the paper.
>
>
> “While the dot product improves the performance significantly, the gain of Laplace is much lower… Do you have any intuition about why performs worse than other models?”
>
> Laplace attention is parameter-free (so keys and queries are just x-values) whereas for dot-product attention, the keys and queries are parameterised representations of the x-values (output of learned MLP that takes x-values as inputs). So the dot-product similarities are computed in a learned representation space, whereas for Laplace the similarities are computed based on L1 distance in the x-space. Hence it is expected that dot-product attention will be able to fit the contexts better than Laplace. We have made this clear in the revised version of the paper.
>
>
> “Do you have any specific application in mind?”
>
> Bayesian Optimisation (BO) is one notable application of (A)NPs since the predictive mean and uncertainty can be used for finding the minimum of a test function drawn from a stochastic process whose realisations can be used to train the (A)NP (toy experiment results shown in Appendix C, referred to in the last paragraph of the section on 1D experiments in main text). The image data experiments also show promise for applying ANPs to arbitrary pixel inpainting, bottom half prediction and mapping images between arbitrary resolutions.

---

> > ### Comment · AnonReviewer1 · 2018-11-25
> > **Thank you for your response**
> >
> > I appreciate the authors' response. All of my questions are resolved. I am looking forward to see the results of the additional experiments.

---

> > ### Comment · AnonReviewer1 · 2018-11-30
> > **The additional results**
> >
> > Dear authors,
> >
> > I wonder whether the authors added the experimental results of the models having self-attention without the cross-attention. If so, could you point out where to look? I could not find it.
> >
> > Thanks,

---

### Official Review · AnonReviewer2 · 2018-11-03
**The authors propose an extension to the recently established framework of Neural Processes by adding an attention-based conditioning mechanism which allows the model to better capture dependencies in its conditioning set.**

**Rating:** 6
**Confidence:** 4

**Review:**

This paper is a joy to review, as it is clearly written and has a crisp idea that the authors try to motivate consistently.
It extends the framework of neural processes and conditional neural processes by an incremental seeming idea: self attention on the conditioning set and cross attention. What this means in practice is that the model is able to learn a more detailed and structured 'kernel' between query and past data which allows it to identify and model conditional structure better.

The authors try three main prongs of such attention mechanisms with the multi-head attention appearing to be the most successful one in the experiments.

Regarding experiments, the authors show a 1d function gitting example and various conditional image generation ones, similar to the original examples in the paper. While I find the function fitting exampole quite unconvin cing, it arguably also contains less interesting structure for the model to pick up.
In the image generation examples both he quantitative and the qualitative illustrations appear to indicate that a very rich conditioning apparatus (stacked multi head attention) manages to give the model more detailed generative abilities.
While introducing all this machinery seems a bit over-engineered at times, the results do show a benefit.

Overall I find the exposition of the effects of the attention mechanism very well executed and the paper clearly positioned and written. My main complaint would be the incremental nature of the work, as the contributions here are not as significant advances as some preceding ideas that have gone into this work, but still steadily improve on the vision of NP and appear to be necessary steps to push the model forward giving this work validity on its own.
The authors discuss a similar mechanism for generation, which while more involved would be a very exciting change from the current framework.  I would have enjoyed seeing more of that in this paper to discuss input and output attention jointly.

---

> ### Author Response · Authors · 2018-11-14
> **Response to AnonReviewer2**
>
> We would like to thank you for your positive review and thoughtful comments. Here are our responses:
>
> “My main complaint would be the incremental nature of the work … but still steadily improve on the vision of NP and appear to be necessary steps to push the model forward giving this work validity on its own.”
>
> Regarding the nature of the work, we show that attention addresses underfitting, which is a fundamental drawback of NPs, so this simple change makes a large difference. We show empirically that attention leads to large improvements in training time, expressiveness of the model, and solves the underfitting issue so that ANPs can be used reliably for tasks such as mapping images from one resolution to another. Hence as you pointed out, we believe that ANPs are a notable improvement to NPs and is valid work in its own right.
>
>
> “The authors discuss a similar mechanism for generation, which while more involved would be a very exciting change from the current framework.”
>
> We gladly agree that the incorporation of self-attention in the decoder is an interesting direction for future research. We are currently investigating this avenue as future work.

---

### Official Review · AnonReviewer3 · 2018-11-05
**This paper proposes to resolve this issue by adding an attention mechanism to the deterministic path.**

**Rating:** 6
**Confidence:** 4

**Review:**

Neural process (NP) is a recent probablistic method for modeling distributions of functions. The authors claim that one substantial weakness of NP is the tendency of under-fitting. The authors give a hypoethesize: the under-fitting behaviour of NP is because the mean-aggregation step in the encoder acts as a bottleneck, as a result, it is difficult for the decoder to learn the relevant information for a give target prediction. This paper proposes to resolve this issue by adding an attention mechanism to the deterministic path. The experimental results show that the proposed method converge faster and give better results on various tasks.

One major concern about the paper is the lack of analysis of the true cause of under-fitting in NP. The authors give the hypoethesize about the potential cause of the under-fitting issue and proposes to resolve it with attention, however, without theoretical or empirical analyses, it is hard to understand the true cause of the under-fitting issue. Although the proposed method give better performance, it is not clear whether the better performance is due to the added complexity to the model (the attention mechanism) or truely resolving the under-fitting issue. Some analyses along this line can make the paper clearer and more convincing.

A lot of technical details are missing in the paper, which makes the method not reproducible. Please add more details about the proposed attention mechanism and how they are implemented into NP.

In the GP literature, there are also methods tackling meta-learning or multi-task learning or few shot learning. These works are known as multi-output / multi-tasks Gaussian processes. A few works on this topic are listed:
* Z Dai, MA Álvarez, ND Lawrence, Efficient Modeling of Latent Information in Supervised Learning using Gaussian Processes, NIPS 2017
* MA Alvarez, L Rosasco, ND Lawrence, Kernels for vector-valued functions: A review, Foundations and Trends® in Machine Learning 2012
* EV Bonilla, KM Chai, C Williams, Multi-task Gaussian process prediction, NIPS 2007

For the 1D regression experiments (Figure 1left, Figure 3right), it is not clear which fitting is better. It largely depends on the prior of kerel parameters. As the data points are generated from a GP, plotting the Gaussian process fit with the ground truth parameters can show what a ground truth fitting would look like.

The Bayesian optimization experiment is very nice and gives some good insights about the quality of the uncertainty of prediction. Maybe consider it to include it in the main text.

---

> ### Author Response · Authors · 2018-11-14
> **Response to AnonReviewer3**
>
> We would like to thank you for your constructive criticism of the paper. Here are our responses:
>
> “The authors give the hypoethesize about the potential cause of the under-fitting issue and proposes to resolve it with attention…. Although the proposed method give better performance, it is not clear whether the better performance is due to the added complexity to the model (the attention mechanism) or truely resolving the under-fitting issue”
>
> Our reasoning that the mean-aggregation acts as a bottleneck is as follows. In NPs, the aggregated representations r_C and s_C needs to have information about the particular function (the realisation of the stochastic process) not only at the context x-values but for all x-values in the domain for the predictions to generalise well to unseen targets. When we compute these aggregated representations simply by taking the mean of representations of each context pair, it will be difficult for the resulting representation to model the entire function, and may require a large representation dimensionality. On the other hand, cross-attention allows the aggregated representation r* to be specific to the target x-value x*, so that it suffices for the representation r* to only be informative about the prediction at x* instead of all x-values in the domain. We claim that this inductive bias allows the ANP to resolve the undefitting issue, not just because of the added model capacity (i.e. more parameters) that attention introduces. As evidence for this, on the left of Figure 3 we show quantitative results for differing sizes of the bottleneck in NPs, ranging from d=128 to 1024. The figure shows that raising d (i.e. increasing model capacity) does help achieve better reconstructions, but there seems to be a limit in how much the reconstructions can improve, hence showing that naively adding capacity to the model is insufficient in addressing the underfitting behaviour. We show that multihead (and dot-product) attention with d=128 gives a much faster decrease in the reconstructions than NPs with any value of d, both against iteration and wall-clock time. We would also like to point out that the number of parameters in the NP model for bottleneck size d is approximately 10d^2 (ignoring smaller order terms), whereas for multihead it is 15d^2 (so cross-attention introduces 5d^2 extra parameters). So the number of parameters of NP with d=1024 is much higher than that of multihead with d=128, yet we can get noticeably better performance with multihead attention. Hence the improvement in performance is mostly due to the attention mechanism rather than the increase in model complexity (i.e. number of parameters).
>
>
> “A lot of technical details are missing in the paper, which makes the method not reproducible.”
>
> In addition to the experimental and model architecture details in Appendix A, we have included Figure 8 detailing the precise architecture for the different models. We believe this is sufficient to reproduce the method, but if you think anything is missing please do let us know and we revise the experimental details accordingly.
>
>
> “In the GP literature, there are also methods tackling meta-learning or multi-task learning or few shot learning.”
>
> Thank you for pointing out these references. We agree that these are relevant work and have added them to the related work section of the paper.
>
>
> “For the 1D regression experiments (Figure 1left, Figure 3right), it is not clear which fitting is better. It largely depends on the prior of kerel parameters. As the data points are generated from a GP, plotting the Gaussian process fit with the ground truth parameters can show what a ground truth fitting would look like.”
>
> In the revised version of the paper, we have included Figure 9 showing the predictive mean and variance of the oracle GP from which the contexts were drawn for comparison in Appendix C. Note that the data was generated from a GP with small observation noise (sigma_n = 0.02), so we do want the predictive mean to pass through the contexts with predictive variance small near the context points and large away from them. It is clear that the Multihead ANP is much closer to the ground truth than the NP with very accurate predictive mean, despite there being evidence of underestimating the variance away from the contexts - one possible explanation for this is that variational inference usually leads to underestimates of predictive variance.

---

### Meta-Review · Area_Chair1 · 2018-12-13
**somewhat limited novelty but good performance**

**Confidence:** 4
**Recommendation:** Accept (Poster)

**Metareview:**

1. Describe the strengths of the paper.  As pointed out by the reviewers and based on your expert opinion.

- The paper is clear and well-motivated.
- The experimental results indicate that the proposed method outperforms the SOTA

2. Describe the weaknesses of the paper. As pointed out by the reviewers and based on your expert opinion. Be sure to indicate which weaknesses are seen as salient for the decision (i.e., potential critical flaws), as opposed to weaknesses that the authors can likely fix in a revision.

- The novelty is somewhat minor.
- An interesting (but not essential) ablation study is missing (but the authors promised to include it in the final version).

3. Discuss any major points of contention. As raised by the authors or reviewers in the discussion, and how these might have influenced the decision. If the authors provide a rebuttal to a potential reviewer concern, it’s a good idea to acknowledge this and note whether it influenced the final decision or not. This makes sure that author responses are addressed adequately.

There were no major points of contention.

4. If consensus was reached, say so. Otherwise, explain what the source of reviewer disagreement was and why the decision on the paper aligns with one set of reviewers or another.

The reviewers reached a consensus that the paper should be accepted.